∂ | Open Peer Review | Clinical Microbiology | Research Article

# The impact of the COVID-19 pandemic on blood culture practices and bloodstream infections

Matt Driedger,[1] Nick Daneman,[2,3] Kevin Brown,[4,5] Wayne L. Gold,[6] Sarah C.J. Jorgensen,[7] Colleen Maxwell,[8] Kevin L. Schwartz,[3,4,5,9] Andrew M. Morris,[10] Deva Thiruchelvam,[3] Bradley Langford,[4,5] Elizabeth Leung,[9] Derek MacFadden[1,3,11]

**ABSTRACT**    The COVID-19 pandemic has likely influenced the epidemiology of bacterial infections through wide-ranging changes to clinical practices and infection control and prevention interventions. We sought to determine how the detection and incidence of bloodstream infections (BSIs) have been influenced by the pandemic. We performed a retrospective analysis of blood culture data in the province of Ontario, Canada, from 1 January 2017 to 31 December 2020. Outcomes included a weekly incidence of blood culture tests, BSIs, and contaminant results. Results were stratified by hospital, community, and long-term care (LTC) settings. An interrupted time series analysis using segmented regression models was used to determine changes in outcome incidence/prevalence during the pre- and peri-pandemic periods. Of the 14,083,853 individuals included, 129,329 (0.92%) developed a bloodstream infection. The blood culture ordering rate increased during the pandemic in the hospital setting only [Incidence rate ratio (IRR) 1.09, 95% confidence interval (CI) 1.01–1.19]. There was a decline in the incidence of community-acquired (IRR 0.95, 95% CI 0.91–0.99) and LTC-acquired (IRR 0.85, 95% CI 0.76–0.94) BSIs. Hospital-acquired BSIs were unchanged. The proportion of blood culture contaminants increased in the community (7% increase, $P < 0.01$) and LTC settings (14% increase, $P < 0.05$). There was decreased incidence of community-acquired *Streptococcus pneumoniae* (IRR 0.43, 95% CI 0.33–0.57) and *Staphylococcus aureus* (IRR 0.91, 95% CI 0.84–0.99) bacteremia. Pandemic-related changes in the performance of blood cultures and the epidemiology of BSIs have implications for current and future pandemic antimicrobial use, healthcare resource allocation, and hospital and laboratory policies.

**IMPORTANCE**    Bacterial infections are a significant cause of morbidity and mortality worldwide. In the wake of the COVID-19 pandemic, previous studies have demonstrated pandemic-related shifts in the epidemiology of bacterial bloodstream infections (BSIs) in the general population and in specific hospital systems. Our study uses a large, comprehensive data set stratified by setting [community, long-term care (LTC), and hospital] to uniquely demonstrate how the effect of the COVID-19 pandemic on BSIs and testing practices varies by healthcare setting. We showed that, while the number of false-positive blood culture results generally increased during the pandemic, this effect did not apply to hospitalized patients. We also found that many infections were likely under-recognized in patients in the community and in LTC, demonstrating the importance of maintaining healthcare for these groups during crises. Last, we found a decrease in infections caused by certain pathogens in the community, suggesting some secondary benefits of pandemic-related public health measures.

**KEYWORDS**    bloodstream infections, pandemic, COVID-19, blood cultures, epidemiology

Address correspondence to Matt Driedger, madriedger@toh.ca.

The authors declare no conflict of interest.

See the funding table on p. 11.

The COVID-19 pandemic has dominated recent efforts within public health and infectious diseases, yet bacterial infection remains a critically important global challenge. Despite the low rate of bacterial co-infection among SARS-CoV-2-infected patients (1, 2), various pandemic-related factors have likely altered the incidence and microbiology of bacterial infections. These include an increased incidence of hospital-acquired bacterial infections, the use of empiric antimicrobials among patients hospitalized with COVID-19 (3–5), patient crowding, and shortages in health human resources and personal protective equipment (PPE) (6). Conversely, mitigating factors may include improved hospital infection prevention and control measures and reduced contact patterns in the community (6).

Large-scale multicenter studies can be used to determine population-level changes in the epidemiology of invasive bacterial infections prior to and during the COVID-19 pandemic. However, existing multicenter studies are generally limited to data from select hospital systems (7–12) and intensive care units (ICUs) (13–16), with varying outcomes perhaps relating to institution-specific factors during the pandemic. Larger nation-wide studies, on the other hand, have generally been limited to hospital-reported nosocomial infections in administrative surveillance data, which may be subject to reporting errors, particularly during the pandemic onset and its associated institutional pressures (15–19). Furthermore, while the pandemic would be expected to have had differential effects on community-acquired infections compared to those acquired in hospitals and other institutions, only a few studies have stratified their analysis accordingly. One recently published large multicenter study in the United States demonstrated an increase in both community- and hospital-acquired bloodstream infections (BSIs) during the first year of the pandemic (20), whereas another analysis demonstrated an increase in only hospital-acquired infections (21). Importantly, both studies utilized only one pre-pandemic year as the comparison period, and so they have limited ability to control for pre-existing trends.

In order to fill these research gaps, we used an interrupted time-series approach to determine changes in the incidence and microbiology of BSIs during the first year of the COVID-19 pandemic in the province of Ontario. We hypothesized that there would be increased blood culture ordering and blood culture contamination in the hospital setting due to staffing shortages and changes in PPE protocols, whereas culture ordering rates would decrease in long-term care (LTC) and community settings due to pandemic-related changes to the provision of healthcare in these settings. We also predicted increased BSI incidence among hospitalized patients and decreased BSI in the community among organisms associated with transmissible infectious syndromes.

## MATERIALS AND METHODS

### Study design and data sources

We performed a retrospective analysis of clinical and microbiology data to determine changes in the incidence rates of bacteremia in the province of Ontario, Canada, prior to and during the first year of the COVID-19 pandemic. Our cohort was derived using data sets at ICES (previously the Institute for Clinical Evaluative Sciences), which were linked using unique encoded identifiers. Data sets included the registered persons database (RPDB), the Canadian Institute for Health Information Discharge Abstract Database (CIHI-DAD), the Continuing Care Reporting System (CCRS), the Ontario Laboratories Information System (OLIS), and others. Linked data sets included demographic and laboratory data from all individuals in the province of Ontario eligible for the provincial health insurance program. Since healthcare in Ontario is publicly funded, this cohort captures information from virtually all 14.5 million residents of Ontario. The use of these data is authorized under Section 45 of Ontario's Personal Health Information Protection Act and does not require research ethics board review (22).

## Inclusion criteria and study period

We analyzed all individuals in Ontario from 1 January 2017 to 31 December 2020. We excluded individuals with invalid birth dates, death dates, or sex. We also excluded individuals who were not Ontario residents or had an invalid ICES linkage number.

## Demographic information

Demographic data including age and sex were extracted, along with clinical data including the Deyo-Charlson Comorbidity Index and recent COVID-19 infection (any microbiologic diagnosis of COVID-19 in the preceding 3 months).

## Outcome definitions

The occurrence of a bloodstream infection was the primary outcome of interest. BSI was defined as blood cultures demonstrating any organism except for *Corynebacterium* spp., *Cutibacterium acnes*, *Micrococcus* spp., or *Bacillus* spp., which were classified as contaminants because they are commonly isolated from the skin and rarely cause BSI. Blood cultures of the same pathogen within a 14-day window were defined as a single BSI episode. Since the growth of coagulase-negative staphylococci (CoNS) is typically interpreted as a contaminant, only sustained CoNS BSI was included as a BSI episode, defined as the same CoNS species detected in two separate blood cultures on separate dates within 14 days. CoNS contaminants, defined as CoNS isolated from cultures for only one day within a 14-day window, were reported separately as a proxy for contamination rate.

Secondary outcomes included BSI by pathogen. All pathogens with sufficient incidence rates to allow interrupted time series (ITS) analysis were included. Common resistance profiles were reported, including methicillin-resistant *Staphylococcus aureus* (MRSA), extended-spectrum beta-lactamase-producing organisms (ESBL), carbapenemase-producing Enterobacterales (CPE), and vancomycin-resistant enterococci (VRE), as determined by susceptibility testing by regional clinical laboratories. Blood culture ordering rates were also expressed as total blood cultures performed per patient-day (or per person for the community cohort).

## Incidence rates

Primary and secondary outcomes were reported in aggregate and by setting. Community-acquired BSI (CBSI) was defined as BSI detected from a blood culture drawn in an ambulatory setting, in the emergency department, or within 2 days of admission; LTC-BSI as blood cultures performed on a patient currently residing in an LTC or within 2 days of admission; and hospital-acquired BSI (HBSI) as detected from blood cultures drawn greater than 2 days after admission. Weekly incidence rates were calculated as a function of total patient days for the hospital and LTC settings and the eligible population (total population minus population in LTC and hospital settings) for community settings. Patient-days and eligible populations used to calculate incidence rates were determined on weekly index dates. Due to reporting limitations on administrative data, incidence rate data were displayed graphically only if they reflected a minimum case count of 5, resulting in the censoring of individual data points from figures (not analyses). For some outcomes with very low rates, monthly incidence rates are shown in figures, but weekly rates were used for underlying analytic models.

## Interrupted time series analysis

Interrupted time series regression models were fitted to detect changes in the incidence of BSI and other outcomes of interest during the pandemic. All ITS models included a binary indicator to determine the change in weekly rates of BSI and other secondary outcomes during the pandemic period (1 March–31 December 2020) compared to the pre-pandemic period (1 January 2017–29 February 2020). 1 March 2020 was selected

as this was the month during which the Ontario government declared a state of emergency requiring the closure of public facilities, masking, and other public health recommendations. We used negative binomial models with log links for rates (blood culture and BSI) and binomial models with logit link for proportions (contaminants). Pre-pandemic temporal trends (slope) and seasonality (where visually present) were incorporated into all-time series in order to control for these variables in determining the pandemic effect. Seasonality was modeled using annual sine and cosine functions to control for seasonality in outcomes and was applied to all time series that visually demonstrated seasonal patterns. Autocorrelation, partial autocorrelation, and residual plots were used to assess model validity. Graphs of weekly or monthly BSI were used to visually examine changes in incidence over time. All statistical analyses were performed using SAS Enterprise Guide version 7.1 (Cary, NC).

## RESULTS

At the start of the historical cohort, the open cohort included 14,084,853 individuals, with 13,992,030 in the community, 75,941 residing in LTCs, and 16,882 who were hospitalized. A total of 129,329 individuals throughout the cohort had BSI, including 7,196 with LTC-BSI, 30,836 with HBSI, and 36,780 with CBSI. Baseline demographic characteristics are shown in Table 1. Age, Deyo-Charlson Comorbidity Index, and sex were comparable before and during the pandemic. Patients with HBSI were more likely to have had a preceding COVID-19 infection compared to those with CBSI (6.8% vs 0.4%, Fisher's exact test $P < 0.001$).

### Blood culture ordering

Figure 1 demonstrates the blood culture ordering rate by setting. In the hospital setting, the weekly blood culture rate increased above pre-pandemic levels (IRR 1.09, 95% CI 1.01–1.19) despite decreased patient volumes (Fig. S1). Blood culture ordering rates in the community and LTC settings were unchanged.

### Probable blood culture contamination

The proportion of CoNS contaminants relative to the weekly blood culture ordering rate increased in the community-acquired [Odds ratio (OR) 1.07, 95% CI 1.02–1.13] and the LTC-acquired settings (OR 1.14, 95% CI 1.01–1.29) (Fig. 2; Table S2). There was a decrease in the proportion of contaminants in the hospital setting (OR 0.90, 95% CI 0.84–0.97).

### BSI incidence

Weekly BSI incidence trends are depicted in Fig. 3A through C. There was no change in the incidence of HBSI (IRR 1.04, 95% CI 0.97–1.11). Conversely, BSI rates decreased at the

**TABLE 1** Baseline characteristics of patients with BSI, by time period and setting[a]

| | | Long-term care | | Hospital | | Community | |
|---|---|---|---|---|---|---|---|
| | | Pre-pandemic | Pandemic | Pre-pandemic | Pandemic | Pre-pandemic | Pandemic |
| | | N = 5,897 | N = 1,299 | N = 23,997 | N = 6,839 | N = 28,560 | N = 8,210 |
| Age, years | Mean ± SD | 81.8 ± 11.2 | 80.6 ± 11.4 | 64.5 ± 19.1 | 63.8 ± 18.0 | 64.6 ± 20.4 | 66.2 ± 19.2 |
| Age group, years | <18 | | | 502 (2.1%) | 101 (1.5%) | 819 (2.9%) | 164 (2.0%) |
| | 18–64 | 487 (8.3%) | 134 (10.3%) | 10,214 (42.6%) | 2,988 (43.7%) | 11,229 (39.3%) | 3,040 (37.0%) |
| | ≥ 65 | 5,410 (91.7%) | 1,165 (89.7%) | 13,281 (55.3%) | 3,750 (54.8%) | 16,512 (57.8%) | 5,006 (61.0%) |
| Deyo-Charlson Index | Mean ± SD | 2.9 ± 2.22 | 2.5 ± 2.1 | 3.5 ± 2.6 | 2.2 ± 2.3 | 2.8 ± 2.6 | 2.1 ± 2.2 |
| Sex | Female | 3,040 (51.6%) | 661 (50.9%) | 10,005 (41.7%) | 2,602 (38.0%) | 12,955 (45.4%) | 3,583 (43.6%) |
| | Male | 2,857 (48.4%) | 638 (49.1%) | 13,992 (58.3%) | 4,237 (62.0%) | 15,605 (54.6%) | 4,627 (56.4%) |
| COVID-19 status | Negative | 5,897 (100.0%) | 1,244 (95.8%) | 23,997 (100.0%) | 6,375 (93.2%) | 28,560 (100.0%) | 8,178 (99.6%) |
| (prior to 90 days) | Positive | 0 (0.0%) | 55 (4.2%) | 0 (0.0%) | 464 (6.8%) | 0 (0.0%) | 32 (0.4%) |

[a]The percentages are column percentages.

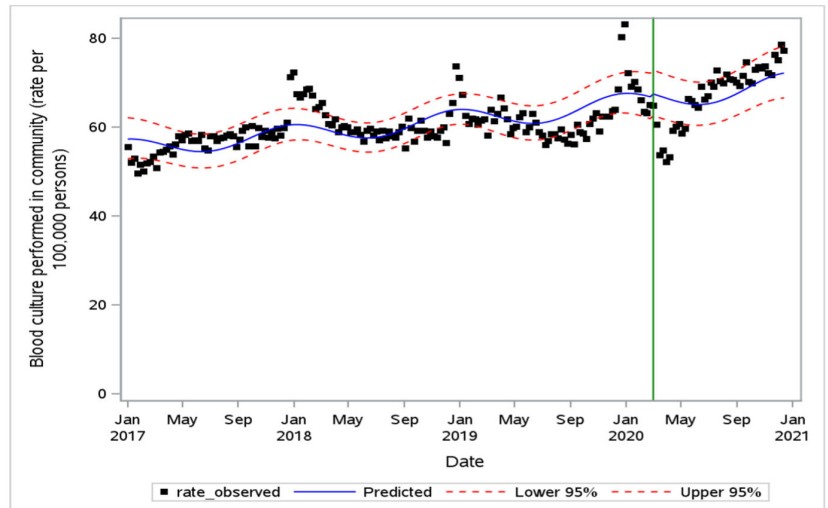

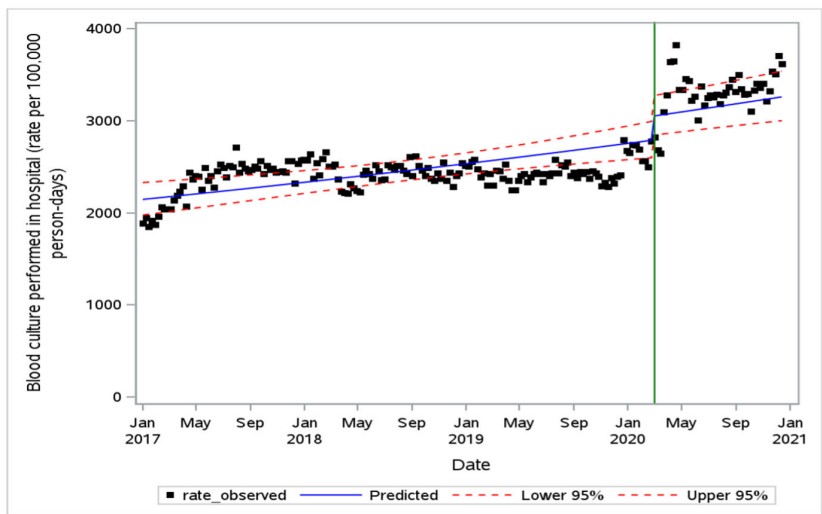

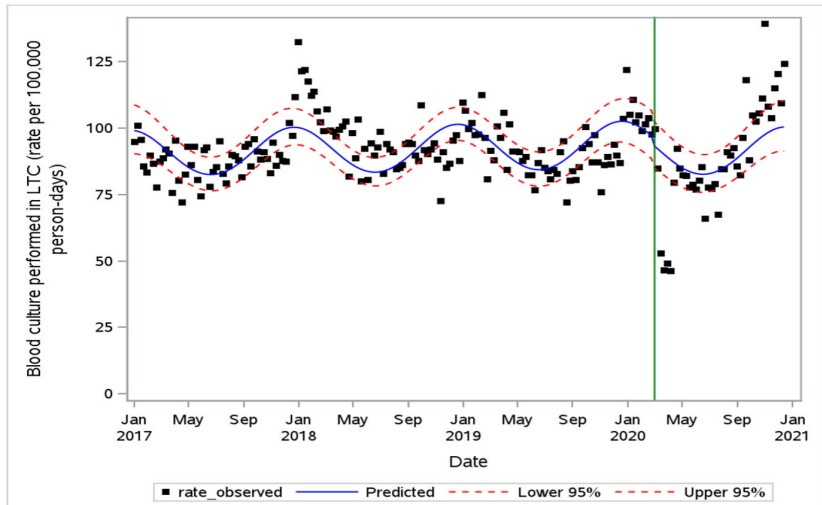

**FIG 1** Weekly blood culture ordering rate in community (A), hospital (B), and LTC (C) settings. The green vertical line indicates the onset of the pandemic period (March 2020).

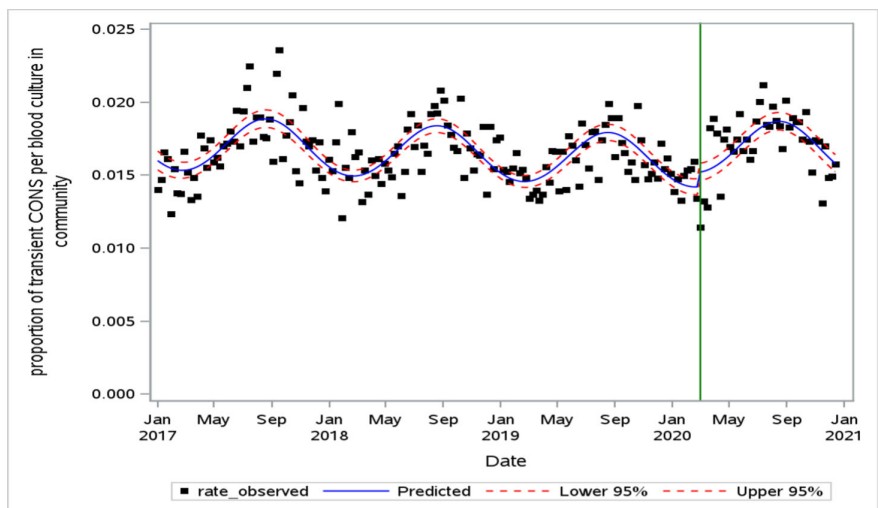

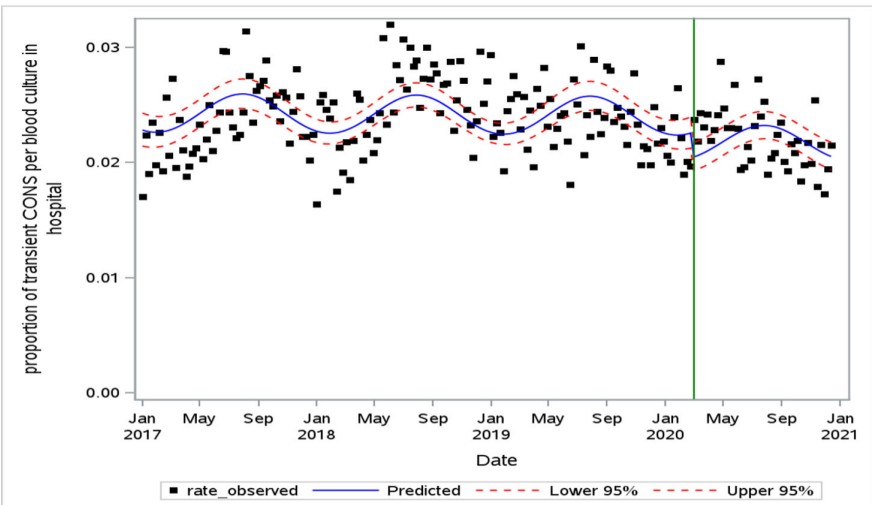

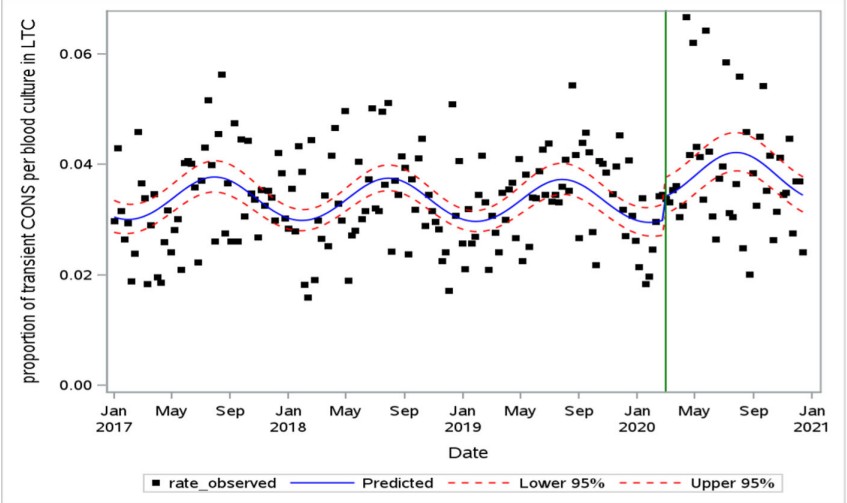

**FIG 2** Weekly proportion of CoNS contaminants per weekly blood cultures in community (A), hospital (B), and LTC (C) settings at one time point censored for low counts (LTC plot). The green vertical line indicates the onset of the pandemic period (March 2020).

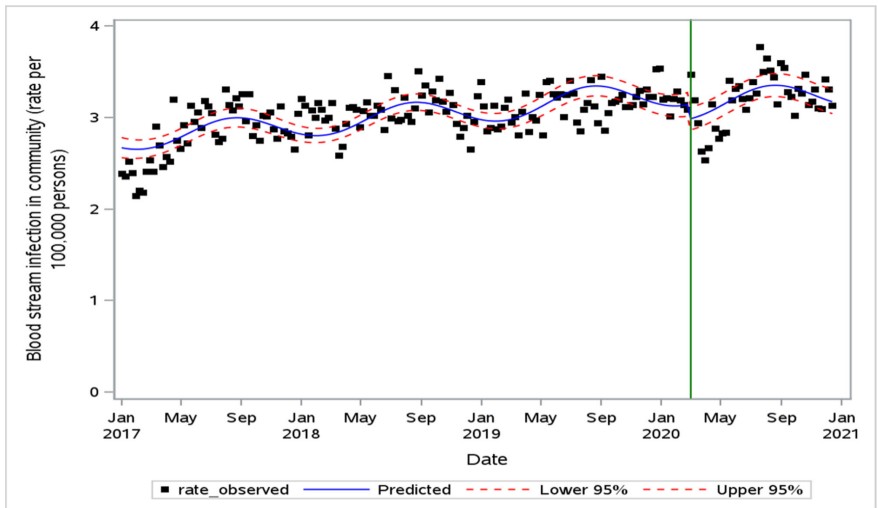

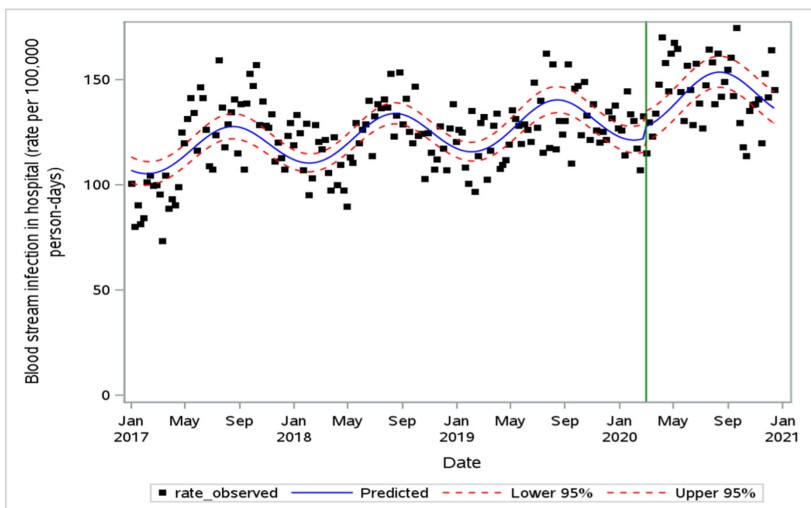

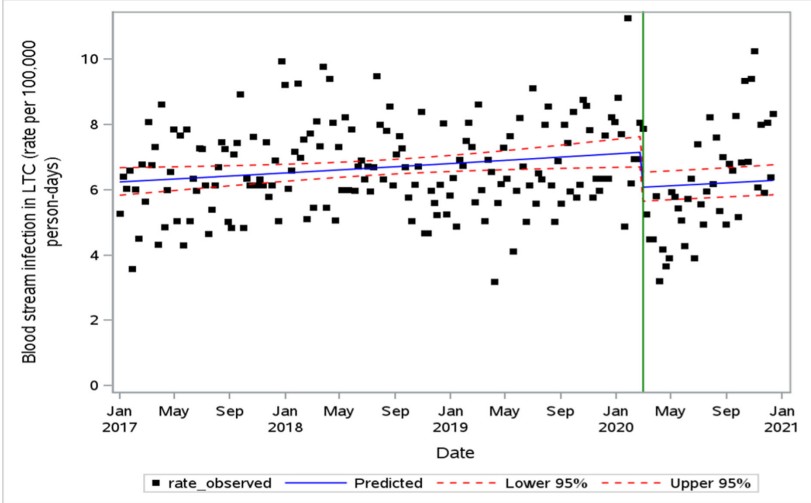

**FIG 3** Weekly BSI incidence in community (A), hospital (B), and LTC (C) settings. The green vertical line indicates the onset of the pandemic period (March 2020).

pandemic onset in community-acquired (IRR 0.95, 95% CI 0.91–0.99) and LTC-acquired (IRR 0.85, 95% CI 0.76–0.94) settings.

## Microbiology of BSI

In the community-acquired setting (Fig. S2; Fig. 2), the pandemic onset was associated with a notable decline in the incidence of *S. pneumoniae* BSI (IRR 0.43, 95% CI 0.33–0.57) and a decline in *S. aureus* BSI (IRR 0.91, 95% CI 0.84–0.99). In the hospital setting (Table 2; Fig. S3), there was an increase in sustained CoNS BSI (IRR 1.24, 95% CI 1.03–1.48) and *Klebsiella* spp. BSI (IRR 1.17, 95% CI 1.04–1.31) figures not shown due to low case counts. Changes in the LTC-acquired setting are shown in Table 2 and Fig. S4. There was a significant decrease in the incidence of LTC-acquired *Escherichia coli* (IRR 0.70, 95% CI 0.52–0.92; figures not shown due to low case counts).

## Resistant organisms

There were no observed differences in the incidence of bacteremia with resistant organisms (MRSA, ESBL, and VRE) in the hospital setting during the pandemic (Table S3; figures not shown due to low case counts). Insufficient incidence rates of BSIs from resistant organisms precluded analysis in community-acquired and LTC-acquired settings.

## Sensitivity analyses

We performed a sensitivity analysis considering the possibility of a post-pandemic slope change (based on visual inspection) in three instances: blood culture ordering rates in the community, blood culture ordering rates in LTCs, and BSI incidence in LTCs (Table S4). While there was a post-pandemic slope change detected in each case, the remainder of the pandemic effect findings were not largely impacted.

## DISCUSSION

In this large province-wide study, we demonstrated that the COVID-19 pandemic onset was associated with changes in the quality and number of blood cultures performed and in the incidence and microbiology of BSIs. There was an increase in the number of blood cultures performed on hospitalized patients. Blood culture contamination increased in the community- and LTC-origin cultures. Overall, BSI incidence decreased in the community and LTC settings but increased for certain pathogens in the hospital setting.

Our cohort, including all blood cultures over 4 years from more than 14 million individuals across various settings, from rural communities to urban referral centers, represents one of the most comprehensive studies to date of the overall incidence of BSI

**TABLE 2** Incidence rate ratios of pathogen-specific BSI in community, hospital, and LTC settings using segmented regression models, comparing peri-pandemic to pre-pandemic periods

| BSI pathogen | Community | | Hospital | | Long-term care | |
|---|---|---|---|---|---|---|
| | IRR[a] | 95% CI[b] | IRR | 95% CI | IRR | 95% CI |
| *S. pneumoniae* | 0.43[e] | 0.33–0.57 | - | - | - | - |
| Group A *Streptococcus* | 0.86 | 0.74–1.01 | - | - | - | - |
| Coagulase-negative staphylococci | 1.18 | 0.95–1.46 | 1.24[d] | 1.03–1.48 | - | - |
| *S. aureus* | 0.91[d] | 0.84–0.99 | 1.01 | 0.90–1.15 | 1.06 | 0.76–1.47 |
| *Enterococcus* spp. | 1.05 | 0.94–1.17 | 1.13 | 1.00–1.28 | 1.14 | 0.74–1.77 |
| *E. coli* | 0.98 | 0.89–1.09 | 0.95 | 0.85–1.08 | 0.70[d] | 0.52–0.92 |
| *Klebsiella* spp. | 1.00 | 0.94–1.09 | 1.17 | 1.04–1.31 | 0.73 | 0.50–1.07 |

[a]Incidence rate ratio.
[b]95% confidence interval.
[c]Analysis not conducted due to insufficient incidence rates.
[d]P value < 0.05.
[e]P value < 0.01.

during the COVID-19 pandemic. Stratifying results by setting of blood culture collection allowed for novel findings, including that community and hospitalized BSI rates were differentially influenced by the pandemic and that the increase in contaminant rates during the pandemic is largely attributable to those cultures performed in emergency department or outpatient settings. The inclusion of data from 3 years of pre-pandemic data allowed us to model each outcome to control for seasonal and temporal trends present prior to the pandemic, which strengthens the validity of our findings given the seasonal variation of bacterial infections.

Smaller cohort studies suggest that while contamination rates increased overall during the pandemic (7, 8), the specific department implicated (ICU, inpatient, or emergency department) varied by institution (13, 23, 24). Similarly, we observed a disproportionate increase in blood cultures ordered in the hospital setting without a change in the proportion of contaminants, whereas there was an increase in the proportion of contaminated cultures in the LTC and community settings, including from outpatient laboratories, emergency departments, and hospitals within 2 days of admission. This may suggest that the various factors theorized to explain the increased contaminant rates during the pandemic (new PPE protocols, increased workload, reassignment of nursing staff to new units) have especially affected emergency departments and ambulatory laboratories rather than admitted patients. Additionally, certain pressures unique to the emergency department, including patient acuity, the backlog of patients due to saturated wards, and the proportion of patients requiring PPE, may have contributed (25). Given that contaminant blood cultures lead to unnecessary empiric antimicrobial use, our findings demonstrate a potential underappreciated cost of the pandemic to patients, including adverse drug events, antimicrobial resistance, and length of stay (26).

We demonstrated no change in hospital-acquired BSI during the pandemic. While this mirrors the findings of smaller studies (14, 27), most cohorts (13, 17, 18, 28, 29) have reported an increase in hospital-acquired bacterial infections. While Ontario faced similar pandemic-related factors as other settings, the decrease in patient volumes in our cohort suggests that crowding may have had a lesser impact on BSI in our setting, which may explain our findings. The increases in gram-positive infections likely relate to increased line-associated infections in hospitals, which may have been balanced by a decreased incidence of other nosocomial bacteremias, perhaps due to factors such as the delay of elective surgeries.

The observed changes in the microbiology of BSI likely represent true epidemiologic shifts during the pandemic. Decreased community-acquired *S. pneumoniae* infections have been described in recent studies (30–32); we identified this same trend in addition to a decline in *S. aureus*. These changes may be due to public health measures leading to a decrease in both the community transmission of these pathogens and of respiratory viruses such as influenza that can result in secondary bacterial pneumonia due to these pathogens (33, 34). In support of this hypothesis, there has recently been an increase in these pathogens attributed to the cessation of public health measures (35, 36). In the hospital setting, the pandemic was associated with an increase in CoNS BSI. Similar trends have been reported in other studies (9, 37) and among SARS-CoV-2-infected patients specifically (3, 4, 37), with explanations including the use of central venous devices, increased line infections due to altered infection prevention protocols to prevent staff exposure, and empiric use of antimicrobials in patients with COVID-19, including third-generation cephalosporins, which may be selected for these organisms (5, 9).

Community- and LTC-onset BSI decreased in our cohort. A proportion of this signal represents a true decrease in the incidence of BSI due to reduced community transmission. However, as a significant proportion of community-acquired BSI events involve pathogens that are typically implicated in "endogenous" infectious disease syndromes (e.g., urinary tract infection) not predominantly transmitted through person-to-person contact, this finding may be attributable to under-detection due to reluctance to seek

care during the pandemic (38). Similarly, the decline in reported LTC-BSI, including Gram-negative organisms such as *E. coli*, may be due to under-detection during the early part of the pandemic as a result of well-documented challenges in meeting care needs in Ontario LTCs during the early pandemic and directives to care for persons in the community due to issues related to hospital capacity (39).

Certain limitations should be acknowledged. First, while patients with bacteremia are typically quite unwell and seek medical attention, the incidence of positive blood cultures would not have captured all true bacteremia episodes, especially as pandemic-related factors may have prevented patients from accessing care and increased palliative supports in LTCs that may have reduced transfers to acute care settings. Second, limiting our analysis to blood culture data allowed us to generally discern true infection among a large data set while avoiding contaminant bacteria from non-sterile cultures. However, individual chart review is required to definitively discern infection from contaminants even among blood cultures, and certain contaminant culture results may have been misclassified as true BSI; this limitation also applies to our classification of persistent and intermittent CoNS BSI. Furthermore, without other corresponding cultures, we were unable to determine the diagnoses underlying BSI (e.g., *S. pneumoniae* BSI secondary to meningitis vs pneumonia). Last, since our analysis was limited to the first year of the pandemic, our findings do not represent the effects of more recent phases of the pandemic.

Our findings serve to highlight certain future priorities. First, our findings related to blood culture contamination emphasize the need for institution-level surveillance and quality improvement initiatives, such as phlebotomy and central venous catheter protocols, to monitor and prevent potentially harmful and costly consequences of care, especially during times of healthcare and public health crises. Second, the potentially undetected BSI in LTCs underscores the need for equitable healthcare to ensure that vulnerable individuals are adequately cared for in a pandemic context. Last, the observed changes in the microbiology and resistance profiles of BSI provide hypotheses for prospective research to better characterize the modifiable factors in clinical care (e.g., empiric antimicrobials) and infection prevention and control measures (e.g., isolation procedures, room sharing) that may underlie our findings and how these can be optimized in advance of future pandemics.

In summary, we found that pandemic-related factors led to changes in the detection, incidence, and microbiology of BSI and that these effects differed by patient setting. Future studies should aim to characterize the specific modifiable factors that underlie the relationship between emerging viral pandemics and the ever-present challenges of invasive bacterial infections and antimicrobial resistance.

## ACKNOWLEDGMENTS

Parts of these materials are based on data and information compiled and provided by the Ontario Ministry of Health (MOH), the Canadian Institute for Health Information (CIHI), Ontario Health (OH), and the Ontario Ministry of Health. This document also used data adapted from the Statistics Canada Postal CodeOM Conversion File, which is based on data licensed from Canada Post Corporation, and/or data adapted from the Ontario Ministry of Health Postal Code Conversion File, which contains data copied under license from Canada Post Corporation and Statistics Canada. The analyses, conclusions, opinions, and statements expressed herein are solely those of the authors and do not reflect those of the funding or data sources; no endorsement is intended or should be inferred.

This work was supported by the Institute for Clinical Evaluative Sciences (ICES), which is funded by an annual grant from the Ontario Ministry of Health (MOH) and the Ministry of Long-Term Care (MLTC). This work was also supported by the Ontario Health Data Platform (OHDP), a Province of Ontario initiative to support Ontario's ongoing response to COVID-19 and its related impacts. This study also received funding from the Canadian Institutes of Health Research. The analyses, conclusions, opinions, and statements expressed herein are solely those of the authors and do not reflect those of the funding

or data sources; no endorsement by ICES, OHDP, the MOH, the MLTC, or CIHI is intended or should be inferred.

All authors contributed to the study concept and design. M.D., D.M., and D.T. performed the data analyses. M.D., D.M., and N.D. drafted the manuscript. All authors contributed to the critical review of the manuscript.

## AUTHOR AFFILIATIONS

[1]Department of Medicine, The University of Ottawa, Ottawa, Ontario, Canada
[2]Sunnybrook Health Sciences Centre, University of Toronto, Toronto, Ontario, Canada
[3]ICES, Toronto, Ontario, Canada
[4]Public Health Ontario, Toronto, Ontario, Canada
[5]Dalla Lana School of Public Health, Toronto, Ontario, Canada
[6]The University Health Network, Toronto, Ontario, Canada
[7]Institute of Medical Science, Toronto, Ontario, Canada
[8]University of Waterloo, Waterloo, Ontario, Canada
[9]Unity Health Toronto, Toronto, Ontario, Canada
[10]Sinai Health System, Toronto, Ontario, Canada
[11]The Ottawa Hospital Research Institute, Ottawa, Ontario, Canada

## AUTHOR ORCIDs

Matt Driedger http://orcid.org/0000-0001-6921-4443
Nick Daneman http://orcid.org/0000-0001-8827-3764
Kevin Brown http://orcid.org/0000-0002-1483-2188
Kevin L. Schwartz http://orcid.org/0000-0002-3666-7005

## FUNDING

| Funder | Grant(s) | Author(s) |
| --- | --- | --- |
| Institute for Clinical Evaluative Sciences (ICES) | | Derek MacFadden |
| Ontario Ministry of Health and Long-Term Care (MOHLTC) | | Derek MacFadden |
| Ontario Health Data Platform | | Derek MacFadden |
| Gouvernement du Canada \| Canadian Institutes of Health Research (IRSC) | | Derek MacFadden |

## AUTHOR CONTRIBUTIONS

Matt Driedger, Conceptualization, Data curation, Formal analysis, Methodology, Writing – original draft, Writing – review and editing | Nick Daneman, Conceptualization, Supervision, Writing – review and editing | Kevin Brown, Conceptualization, Writing – review and editing | Wayne L. Gold, Conceptualization, Writing – review and editing | Sarah C.J. Jorgensen, Conceptualization, Writing – review and editing | Colleen Maxwell, Conceptualization, Writing – review and editing | Kevin L. Schwartz, Conceptualization, Writing – review and editing | Andrew M. Morris, Conceptualization, Writing – review and editing | Deva Thiruchelvam, Conceptualization, Data curation, Formal analysis, Writing – review and editing | Bradley Langford, Conceptualization, Writing – review and editing | Elizabeth Leung, Conceptualization, Writing – review and editing | Derek MacFadden, Conceptualization, Data curation, Methodology, Supervision, Writing – review and editing

## DATA AVAILABILITY

Data was provided and stored by MOH and CIHI as above. The dataset from this study is held securely in coded form at ICES. While legal data sharing agreements between ICES and data providers (e.g., healthcare organizations and government) prohibit ICES from making the dataset publicly available, access may be granted to those who meet

pre-specified criteria for confidential access, available at www.ices.on.ca/DAS (email: das@ices.on.ca). The full dataset creation plan and underlying analytic code are available from the authors upon request, understanding that the computer programs may rely upon coding templates or macros that are unique to ICES and are therefore either inaccessible or may require modification.

## ADDITIONAL FILES

The following material is available online.

### Supplemental Material

**Supplemental material (Spectrum02630-23-s0001.docx).** Tables S1 to S4; Fig. S1 to S4.

### Open Peer Review

**PEER REVIEW HISTORY (review-history.pdf).** An accounting of the reviewer comments and feedback.

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
