## [Reviewer comments · Microbiology Spectrum]

Microbiology Spectrum

The Impact of the COVID-19 Pandemic on Blood Culture Practices and Bloodstream Infections

Matthew Driedger, Nick Daneman, Kevin Brown, Wayne Gold, Sarah Jorgensen, Colleen Maxwell, Kevin Schwartz, Andrew Morris, Deva Thiruchelvam, Bradley Langford, Elizabeth Leung, and Derek MacFadden

Corresponding Author(s): Matthew Driedger, University of Ottawa

Review Timeline:

Submission Date:	June 26, 2023
Editorial Decision:	July 12, 2023
Revision Received:	September 8, 2023
Accepted:	October 11, 2023

Editor: Ahmed Babiker

Reviewer(s): The reviewers have opted to remain anonymous.

Transaction Report:

DOI: <https://doi.org/10.1128/spectrum.02630-23>

July 12, 2023

Dr. Matthew Driedger
University of Ottawa
Medicine
501 Smyth Rd
Ottawa K1H 8L6
Canada

Re: Spectrum02630-23 (The Impact of the COVID-19 Pandemic on Blood Culture Practices and Bloodstream Infections)

Dear Dr. Matthew Driedger:

Thank you for submitting your manuscript to Microbiology Spectrum. Your article has been reviewed by two reviewers who raise some important points.

Link Not Available

Sincerely,

Ahmed Babiker

Journals Department
Reviewer comments:

Reviewer #1 (Comments for the Author):

The authors conducted a retrospective review of blood culture and administrative data to analyze the impact of the early COVID-19 pandemic (March - December 2020) on the incidence of blood culture collections, blood stream infections, and various secondary outcomes in Ontario, Canada. They use a level-change interrupted time series (ITS) analysis to make these comparisons. They report observed differences in the rate of blood culture collection in hospitalized patients (increased), the rate

of CoNS blood culture contamination in community/LTC cultures (decreased), and in the rate of blood stream infection detection in community/LTC (decreased). In general, the observed effect sizes are small. The manuscript is clearly written.

Major Points

1. Choice of Impact Model

The authors do not provide an explanation for their choice of ITS impact model. While looking for a level-change makes intuitive sense, there are additional considerations when considering the impact of the pandemic on the studied outcomes. Some changes due to the COVID-19 pandemic such as staffing/supply challenges may not be expected necessarily to manifest immediately and might be best modeled with a slope-change model and/or a lag-change model (see Bernal 2017 PMID 27283160) or even a combination of level and slope change models. For example, there visually appears to be a difference in the blood culture ordering rate between the pre- and peri-pandemic periods in the community setting with a transient dip occurring at the time of the declaration of the pandemic. A level + slope change model could be used to determine if this increase is the result of a pre-existing trend or if the onset of the pandemic may have contributed.

2. Inclusion of summary statistics/Interpretation of ITS

The authors use of a rigorous statistical method (ITS) is commendable. However, there are some instances where I think the results are misapplied and a "zoomed out" perspective might be instructive. For example, in Figure 3B it appears that the peri-pandemic hospital BSI rate is higher than the pre-pandemic rate. However, this could be due to continuation of a pre-pandemic trend based on simple visual review of the data. The authors claim, based on the ITS analysis, that there was no change in hospital BSI but this visually seems to not be the case. It would be informative to see the median pre- and peri-pandemic hospital BSI rates compared. ITS (perhaps with a level + slope change impact model, see above) could then be used to determine how likely any observed difference is to be attributable to the pandemic. In short, a non-significant ITS co-efficient does not mean that there is no difference, just that the data does not support an immediate effect (in a level change model) of the onset of the pandemic. Figure 1A appears to be another example of this.

A related example is Figure 3A (community BSI). Here visual inspection shows higher BSI rates in the peri-pandemic period compared to pre-pandemic. Inclusion of summary statistics would confirm this. However, the ITS model returned a significant coefficient for a level change and so a significant decrease in community BSI is reported by the authors. This is likely driven by the small cluster of datapoints in March/April 2020 that are substantially below the overall trend.

3. Determination of blood culture contamination

The authors deal with suspected blood culture contamination in two separate ways without clear explanation as to why. All blood cultures with *Corynebacterium* spp., *Cutibacterium acnes*, *Micrococcus* spp., or *Bacillus* spp. were excluded. Conversely, CoNS identified only on one day in a 14-day period was used as proxy for contamination whereas if the same CoNS species was identified more than once in a 14-day period it was treated as a true BSI. If the data is available, including the other skin flora in the calculation of contamination rate would give a more complete picture.

4. Limitations

The authors discuss several important limitations of their work including concerns about missing data particularly in regards to LTC patients. Limitations also include the inability to identify which aspects of the pandemic and the response to it might have contributed to the observed changes. This affects the impact of these observations in terms of their ability to direct future research.

Minor Points

1. Supplemental Figure 2- Panel A is listed as being by week but data points appear to be graphed by month. Conversely, Panel B appears to be by week but is labeled as by month.

Reviewer #2 (Comments for the Author):

This is a large study involving at the population-level of a province in Canada. The method of analysis using segmented regression models and stratified by hospital, community, and long-term care settings is adapted.

Only the first 9 months of the pandemic were taken into account. It would have been interesting to include the year 2021, especially as we are already more than 3 years from the start of the pandemic. Moreover, there are already many articles on the impact of the COVID-19 pandemic on BSIs, including reviews, and the article presented does not really add anything new.

There are several methodological considerations:

-The time taken to differentiate between nosocomial and community-acquired is 48 hours in international definitions, not 72 hours.

- The definition of CoNS BSI does not match international definitions either. Without clinical data (initiation of antibiotherapy, etc.), it is difficult to conclude that it is a CoNS infection.
- The number of patients in the community, in LTCs, and in hospitals are specified. These data evolve day by day. In addition, the date considered to be the beginning of the pandemic period is not specified.
- The cohort is large enough to present only significant results. The terms "did not reach statistical significance" or "trend" or "statistical power limited" should not be used.

In addition, the results section is short compared to the methods and discussion sections, while many results are presented in tables and figures.

The results show significantly fewer contaminants in the hospital and more in the community and LTC. The proposed explanation ("attributable to new PPE protocols, increased workload, reassignment of nursing staff to new units, or other factors that primarily affected emergency departments and ambulatory laboratories") is also valid for the hospital, but does not explain why a decrease is observed there.

Line 53: "the effects of the pandemic on bacterial infections remains to be established". This is not correct, the existing literature should be cited and the contribution of the article to this literature should be specified.

Line 59: "non-transmissible infections were likely under-recognized". The definition of "non-transmissible infections" is not clear, and the choice of the term "non-transmissible" does not seem appropriate, since infections are caused by transmissible germs.

Line 57: "blood tests for infection were more likely represent false-positive results, which likely cause unnecessarily antimicrobials and hospital stays for patients". Contaminant in an isolated blood culture does not lead to a prescription.

Bacterial names should appear in italics in the figures.

Staff Comments:

Preparing Revision Guidelines

Please return the manuscript within 60 days; if you cannot complete the modification within this time period, please contact me. If you do not wish to modify the manuscript and prefer to submit it to another journal, please notify me of your decision immediately so that the manuscript may be formally withdrawn from consideration by Microbiology Spectrum.

The authors conducted a retrospective review of blood culture and administrative data to analyze the impact of the early COVID-19 pandemic (March – December 2020) on the incidence of blood culture collections, blood stream infections, and various secondary outcomes in Ontario, Canada. They use a level-change interrupted time series (ITS) analysis to make these comparisons. They report observed differences in the rate of blood culture collection in hospitalized patients (increased), the rate of CoNS blood culture contamination in community/LTC cultures (decreased), and in the rate of blood stream infection detection in community/LTC (decreased). In general, the observed effect sizes are small. The manuscript is clearly written.

Major Points

1. Choice of Impact Model

The authors do not provide an explanation for their choice of ITS impact model. While looking for a level-change makes intuitive sense, there are additional considerations when considering the impact of the pandemic on the studied outcomes. Some changes due to the COVID-19 pandemic such as staffing/supply challenges may not be expected necessarily to manifest immediately and might be best modeled with a slope-change model and/or a lag-change model (see Bernal 2017 PMID 27283160) or even a combination of level and slope change models. For example, there visually appears to be a difference in the blood culture ordering rate between the pre- and peri-pandemic periods in the community setting with a transient dip occurring at the time of the declaration of the pandemic. A level + slope change model could be used to determine if this increase is the result of a pre-existing trend or if the onset of the pandemic may have contributed.

2. Inclusion of summary statistics/Interpretation of ITS

The authors use of a rigorous statistical method (ITS) is commendable. However, there are some instances where I think the results are misapplied and a “zoomed out” perspective might be instructive. For example, in Figure 3B it appears that the peri-pandemic hospital BSI rate is higher than the pre-pandemic rate. However, this could be due to continuation of a pre-pandemic trend based on simple visual review of the data. The authors claim, based on the ITS analysis, that there was no change in hospital BSI but this visually seems to not be the case. It would be informative to see the median pre- and peri-pandemic hospital BSI rates compared. ITS (perhaps with a level + slope change impact model, see above) could then be used to determine how likely any observed difference is to be attributable to the pandemic. In short, a non-significant ITS co-efficient does not mean that there is no difference, just that the data does not support an immediate effect (in a level change model) of the onset of the pandemic. Figure 1A appears to be another example of this.

A related example is Figure 3A (community BSI). Here visual inspection shows higher BSI rates in the peri-pandemic period compared to pre-pandemic. Inclusion of summary statistics would confirm this. However, the ITS model returned a significant coefficient for a level change and so a significant decrease in community BSI is reported by the authors. This is likely driven by the small cluster of datapoints in March/April 2020 that are substantially below the overall trend.

3. Determination of blood culture contamination

The authors deal with suspected blood culture contamination in two separate ways without clear explanation as to why. All blood cultures with *Corynebacterium* spp., *Cutibacterium acnes*, *Micrococcus* spp., or *Bacillus* spp. were excluded. Conversely, CoNS identified only on one day in a 14-day period was used as proxy for contamination whereas if the same CoNS species was identified more than once in a 14-day period it was treated as a true BSI. If the data is available, including the other skin flora in the calculation of contamination rate would give a more complete picture.

4. Limitations

The authors discuss several important limitations of their work including concerns about missing data particularly in regards to LTC patients. Limitations also include the inability to identify which aspects of the pandemic and the response to it might have contributed to the observed changes. This affects the impact of these observations in terms of their ability to direct future research.

Minor Points

1. Supplemental Figure 2- Panel A is listed as being by week but data points appear to be graphed by month. Conversely, Panel B appears to be by week but is labeled as by month.

This is a large study involving at the population-level of a province in Canada. The method of analysis using segmented regression models and stratified by hospital, community, and long-term care settings is adapted.

Only the first 9 months of the pandemic were taken into account. It would have been interesting to include the year 2021, especially as we are already more than 3 years from the start of the pandemic. Moreover, there are already many articles on the impact of the COVID-19 pandemic on BSIs, including reviews, and the article presented does not really add anything new.

There are a several methodological considerations:

-The time taken to differentiate between nosocomial and community-acquired is 48 hours in international definitions, not 72 hours.

-The definition of CoNS BSI does not match international definitions either. Without clinical data (initiation of antibiotherapy, etc.), it is difficult to conclude to conclude that it is a CoNS infection.

-The number of patients in the community, in LTCs, and in hospitals are specified. These data evolve day by day. In addition, the date considered to be the beginning of the pandemic period is not specified.

-The cohort is large enough to present only significant results. The terms "did not reach statistical significance" or "trend" or "statistical power limited" should not be used.

In addition, the results section is short compared to the methods and discussion sections, while many results are presented in tables and figures.

The results show significantly fewer contaminants in the hospital and more in the community and LTC. The proposed explanation ("attributable to new PPE protocols, increased workload, reassignment of nursing staff to new units, or other factors that primarily affected emergency departments and ambulatory laboratories") is also valid for the hospital, but does not explain why a decrease is observed there.

Line 53: "the effects of the pandemic on bacterial infections remains to be established". This is not correct, the existing literature should be cited and the contribution of the article to this literature should be specified.

Line 59: "non-transmissible infections were likely under-recognized". The definition of "non-transmissible infections" is not clear, and the choice of the term "non-transmissible" does not seem appropriate, since infections are caused by transmissible germs.

Line 57: "blood tests for infection were more likely represent false-positive results, which likely cause unnecessarily antimicrobials and hospital stays for patients". Contaminant in an isolated blood culture does not lead to a prescription.

Bacterial names should appear in italics in the figures.

Response to Reviewer Comments

September 4, 2023

Ahmed Babiker, MD, MSc, MBBS
Editor
Microbiology Spectrum

Dear Dr. Ahmed Babiker,

Thank you for the insightful comments offered by the reviewers of our manuscript, *The Impact of the COVID-19 Pandemic on Blood Culture Practices and Bloodstream Infections: A Time Series Analysis*.

All comments have been addressed. Please find below a point-by-point response to each comment.

We also note that the Research Articles format suggests placing the methods section at the end of the manuscript. We are happy to reformat our manuscript accordingly, or to have editors make this change on our behalf.

We thank you for considering our manuscript, and look forward to hearing from you.

Your Sincerely,

Matt Driedger, MD
Department of Medicine
University of Ottawa
madriedger@toh.ca

Reviewer #1 Comments

The authors conducted a retrospective review of blood culture and administrative data to analyze the impact of the early COVID-19 pandemic (March - December 2020) on the incidence of blood culture collections, blood stream infections, and various secondary outcomes in Ontario, Canada. They use a level-change interrupted time series (ITS) analysis to make these comparisons. They report observed differences in the rate of blood culture collection in hospitalized patients (increased), the rate of CoNS blood culture contamination in community/LTC cultures (decreased), and in the rate of blood stream infection detection in community/LTC (decreased). In general, the observed effect sizes are small. The manuscript is clearly written.

Major Points

1) Choice of Impact Model

The authors do not provide an explanation for their choice of ITS impact model. While looking for a level-change makes intuitive sense, there are additional considerations when considering the impact of the pandemic on the studied outcomes. Some changes due to the COVID-19 pandemic such as staffing/supply challenges may not be expected necessarily to manifest immediately and might be best modeled with a slope-change model and/or a lag-change model (see Bernal 2017 PMID 27283160) or even a combination of level and slope change models.

Response: We specified our ITS impact model based upon the impacts we expected, which we hypothesized would most consistently be step changes. While we agree that there are theoretical bases for using different models (addition of slope or lag) to account for the changing effects of the pandemic over time, (impact on staffing, supply challenges are two among many) we feel there are more impactful parameters that have an immediate effect (eg. public health closures, changes in PPE use in hospitals, changes in care seeking behaviours). Ultimately, since we could not predict *a priori* whether the culmination of these pandemic-related factors would occur immediately or over time, a level change impact model was decided upon for simplicity.

For example, there visually appears to be a difference in the blood culture ordering rate between the pre- and peri-pandemic periods in the community setting with a transient dip occurring at the time of the declaration of the pandemic. A level + slope change model could be used to determine if this increase is the result of a pre-existing trend or if the onset of the pandemic may have contributed.

Response: We are cautious to apply additional models/statistical testing to our data, as we feel the step change is the most representative response model to our anticipated impacts. However, we have now applied a level + slope analysis for selected models that visually appeared to demonstrate slope changes. These include Figure 1A, Figure 1C and Figure 3C. These results are included as sensitivity analyses in the supplementary materials (Supplementary Table S4) and are referenced in the body of the manuscript (page 10, lines 14-19).

2. Inclusion of summary statistics/Interpretation of ITS

The authors use of a rigorous statistical method (ITS) is commendable. However, there are some instances where I think the results are misapplied and a "zoomed out" perspective might be instructive. For example, in Figure 3B it appears that the peri-pandemic hospital BSI rate is higher than the pre-pandemic rate. However, this could be due to continuation of a pre-pandemic trend based on simple visual review of the data. The authors claim, based on the ITS analysis, that there was no change in hospital BSI but this visually seems to not be the case. It would be informative to see the median pre- and peri-pandemic hospital BSI rates compared.

ITS (perhaps with a level + slope change impact model, see above) could then be used to determine how likely any observed difference is to be attributable to the pandemic. In short, a non-significant ITS co-efficient does not mean that there is no difference, just that the data does not support an immediate effect (in a level change model) of the onset of the pandemic. Figure 1A appears to be another example of this.

A related example is Figure 3A (community BSI). Here visual inspection shows higher BSI rates in the peri-pandemic period compared to pre-pandemic. Inclusion of summary statistics would confirm this. However, the ITS model returned a significant coefficient for a level change and so a significant decrease in community BSI is reported by the authors. This is likely driven by the small cluster of datapoints in March/April 2020 that are substantially below the overall trend.

Response: The included model does account for overall temporal trends (i.e., pre-existing slope and seasonality are variables within the model) in the years prior to (and after) the pandemic, in determining whether the onset of the pandemic was associated with a step change in overall incidence rates or proportions. Figures 3A and 3B, highlighted by the reviewer, are actually an example of this. In Figure 3B, the model (depicted by the blue line) does demonstrate a pre-existing increase in BSI rates prior to the pandemic, and so the increased rates following the pandemic onset (as noted by the reviewer) are, according to the model, a reflection of this pre-existing trend, rather than a result of the pandemic effect. Therefore, there is no significant pandemic effect using a step-change assumption. Inclusion of the median incidence rates during the pre-pandemic versus peri-pandemic periods would not necessarily distinguish between a pre-existing trend and a true pandemic effect. It is only where changes in the outcome cannot be predicted by the model (i.e., are not a "pre-existing trend") that a significant pandemic effect exists (e.g., Figure 3A). Further clarification of this concept has been added (page 8, lines 10-12).

3. Determination of blood culture contamination

The authors deal with suspected blood culture contamination in two separate ways without clear explanation as to why. All blood cultures with *Corynebacterium* spp., *Cutibacterium acnes*, *Micrococcus* spp., or *Bacillus* spp. were excluded. Conversely, CoNS identified only on one day in a 14-day period was used as proxy for

contamination whereas if the same CoNS species was identified more than once in a 14-day period it was treated as a true BSI. If the data is available, including the other skin flora in the calculation of contamination rate would give a more complete picture.

Response: Since coagulase-negative staphylococci collectively represent the most common isolates for contaminant result, we felt that using CoNS as a proxy for contamination rate would be sufficient for our research question, that is, whether there was a change in proportion of contaminants. Since accurately differentiating contaminant from true BSI requires in-depth chart review that was not feasible given our large dataset, we did not aim to discern the precise proportion of contaminant cultures in general, which is well-published elsewhere. This was added as a limitation (page 13, lines 22-25). A similar approach has been used in other studies, including those cited in our paper (e.g., Denny et al., ref. #7).

Minor Points

1. Supplemental Figure 2- Panel A is listed as being by week but data points appear to be graphed by month. Conversely, Panel B appears to be by week but is labeled as by month.

Response: We have corrected this error in title labeling (page 23).

Reviewer #2 Comments

This is a large study involving at the population-level of a province in Canada. The method of analysis using segmented regression models and stratified by hospital, community, and long-term care settings is adapted.

Only the first 9 months of the pandemic were taken into account. It would have been interesting to include the year 2021, especially as we are already more than 3 years from the start of the pandemic.

Response: We agree. This limitation has been highlighted in our manuscript (page 14, lines 2-3).

Moreover, there are already many articles on the impact of the COVID-19 pandemic on BSIs, including reviews, and the article presented does not really add anything new.

Response: Additional multi-centre studies have been published since our last literature review, and these have been added to the manuscript and referenced. However, our study remains an important contribution to the body of literature on this topic. Our study is among the most comprehensive population-level studies, and is unique in its reporting of

the distinct effects on individuals in the community, hospital, and long-term care setting, acknowledging that the effects of the pandemic would be expected to have had varying effects by setting. The two studies of comparable sample size that include both community- and hospital-onset infections (Bauer et al. [ref. #20], Sturm et al. [ref. #21]) include data from only one year prior to the pandemic for comparison; consequently, pandemic effects were determined by simple statistical tests that cannot account for pre-existing temporal and seasonal trends. This limitation applies to the majority of multi-centre studies on this topic. This is in contrast to the modeling method in our study, which utilizes trends from 3 years of pre-pandemic data. The remaining multi-centre studies of comparable size to our work (eg. Weiner-Lastinger et al. [ref. #17], Evans et al. [ref. #18], Baker et al [ref. #19]) include only data on nosocomial infections using national administrative datasets that may be limited by under-reporting particularly during the time of the pandemic. A more in-depth overview of published work has been added in the introduction section (page 4, lines 13-25; page 5, lines 1-2). The particular strengths of our study and its contribution to the literature has been added in the discussion section page 11, lines 5-14) as well as in the “Importance” section of the abstract, which has been re-structured (page 3, lines 2-12).

There are a several methodological considerations:

-The time taken to differentiate between nosocomial and community-acquired is 48 hours in international definitions, not 72 hours.

Response: Our dataset included only calendar dates of admission and blood culture collection, and did not include exact times. Therefore, in order to avoid misclassifying community-acquired infections as nosocomial (ie. those occurring two days after admission but less than 48 hours), a 72-hour definition was chosen. This same definition is used in other cohorts on this topic, which are referenced in our manuscript. We have adjusted the terminology in our manuscript to reflect our definition more precisely (ie. changed “72 hours” to “within 2 days of admission”).

-The definition of CoNS BSI does not match international definitions either. Without clinical data (initiation of antibiotherapy, etc.), it is difficult to conclude to conclude that it is a CoNS infection.

Response: We agree that CoNS bacteremia was unlikely to have been classified as contaminant versus infection with perfect accuracy. However, for the purpose of a population-based study without chart review, repeat positive cultures is the best metric available to differentiate infection from contamination. Other studies cited in our manuscript use a similar approach. Our data did not include information on antimicrobial use and our use of large datasets precluded individual chart review. This has been added as a limitation (page 13, lines 22-25).

-The number of patients in the community, in LTCs, and in hospitals are specified. These data evolve day by day. In addition, the date considered to be the beginning of the pandemic period is not specified.

Response: The denominator (population for community, patient-days for hospital/LTC) was recorded on a weekly basis and used to calculate each weekly incidence rate. This effectively controlled for changes in sample size during our study. March 1 was the precise start date of the pandemic period in our analysis, and we have added precise dates to the text for clarity (page 8, lines 5-6).

-The cohort is large enough to present only significant results. The terms "did not reach statistical significance" or "trend" or "statistical power limited" should not be used.

Response: We agree that our cohort is large enough to provide adequate power for our main research questions. Reference to other secondary results that were not statistically significant have been removed (page 10, lines 1, 4-5; page 12, line 19).

In addition, the results section is short compared to the methods and discussion sections, while many results are presented in tables and figures.

Response: We have a large number of tables and figures, which efficiently present many of the results. We have chosen not to extend the text of the results section accordingly, apart from key summaries, as it would extend the length of the manuscript without adding to the content.

The results show significantly fewer contaminants in the hospital and more in the community and LTC. The proposed explanation ("attributable to new PPE protocols, increased workload, reassignment of nursing staff to new units, or other factors that primarily affected emergency departments and ambulatory laboratories") is also valid for the hospital, but does not explain why a decrease is observed there.

Response: We agree that many of these factors may apply to clinical areas more broadly. However, the extent to which these factors differentially influence different healthcare environments is unclear, and our findings may suggest that these factors apply especially to emergency departments. Certain factors are also unique to emergency departments as well. We have amended this section for clarity (page 11, lines 22-25; page 12, lines 1-2).

Line 53: "the effects of the pandemic on bacterial infections remains to be established". This is not correct, the existing literature should be cited and the contribution of the article to this literature should be specified.

Response: We agree – we have tempered this statement in the Importance paragraph. We have generally re-structured this paragraph (page 3, lines 2-12) to more clearly highlight the specific contribution of our article to the literature. Please refer to our response to the second comment by reviewer #2 for further details on the contribution of our article to the literature.

Line 59: "non-transmissible infections were likely under-recognized". The definition of "non-transmissible infections" is not clear, and the choice of the term "non-

transmissible" does not seem appropriate, since infections are caused by transmissible germs.

Response: We agree that this statement is unclear without further explanation, which is difficult to add due to abstract word count limitations. We have removed reference to “transmissible” versus “non-transmissible” pathogens from the abstract. This concept is described more clearly in the body of the manuscript (page 13, lines 7-9).

Line 57: "blood tests for infection were more likely represent false-positive results, which likely cause unnecessarily antimicrobials and hospital stays for patients".
Contaminant in an isolated blood culture does not lead to a prescription.

Response: We agree that isolated blood culture results should not lead to antimicrobial prescriptions. However, unfortunately, existing evidence suggests that antimicrobials are often erroneously prescribed in response to contaminant results. This is referenced in the body of the manuscript (page 12, lines 3-5) In order to accommodate the abstract word count, this point has been removed from the Importance section (page 3, line 9), but remains in the body of the manuscript.

Bacterial names should appear in italics in the figures.

Response: Thank you. We have italicized all pathogen names where appropriate.

October 11, 2023

Dr. Matthew Driedger
University of Ottawa
Medicine
501 Smyth Rd
Ottawa K1H 8L6
Canada

Re: Spectrum02630-23R1 (The Impact of the COVID-19 Pandemic on Blood Culture Practices and Bloodstream Infections)

Dear Dr. Matthew Driedger:

Your manuscript has been accepted, and I am forwarding it to the ASM Journals Department for publication. You will be notified when your proofs are ready to be viewed.

Sincerely,

Ahmed Babiker
Editor, Microbiology Spectrum
